# A Combination of Long-Duration Electrical Stimulation with External Shoulder Support during Routine Daily Activities in Patients with Post-Hemiplegic Shoulder Subluxation: A Randomized Controlled Study

**DOI:** 10.3390/ijerph19159765

**Published:** 2022-08-08

**Authors:** Chen Lavi, Michal Elboim-Gabyzon, Yuval Naveh, Leonid Kalichman

**Affiliations:** 1Department of Rehabilitation, Bait-Balev Hospital, Bat-Yam 59315, Israel; 2Department of Physical Therapy, Faculty of Social Welfare and Health Sciences, University of Haifa, Haifa 3498838, Israel; 3Department of Physical Therapy, Recanati School for Community Health Professions, Ben-Gurion University of the Negev, Beer Sheva 8410501, Israel

**Keywords:** stroke, shoulder subluxation, neuromuscular electrical stimulation

## Abstract

The study objective was to determine the effect of long-duration neuromuscular electric stimulation (NMES) on shoulder subluxation and upper-extremity function during the acute post-stroke stage. Twenty-eight subjects (mean age ± standard deviation −70.0 ± 14.0 years) were randomly assigned to an experimental or to a control group receiving NMES to the supraspinatus and posterior deltoid muscles or sham treatment for 6 weeks. All the subjects continued standard rehabilitation and external shoulder support (EST). Assessments were conducted pre- and post-intervention and at a 2 week follow-up session by an assessor blind to group allocation. Outcome measures included the degree of shoulder subluxation, Fugl–Meyer assessment-upper extremity (FMA-UE) test, FMA—hand and finger subscales, Functional Independence Measure (FIM), and shoulder pain (using the Numeric Pain Rate Scale). Shoulder subluxation was significantly lower, while the FMA-UE and FMA—hand and finger subscales were significantly improved in the experimental group post-intervention and at follow-up compared to the control group. FIM at follow-up improved more in the experimental group. No change was observed in pain level in both groups. Supplementing NMES to standard rehabilitation and EST is beneficial in reducing shoulder subluxation and improving upper-extremity function. Further research is necessary to determine effect of longer treatment duration and longer follow-up periods.

## 1. Introduction

Shoulder subluxation appears in 15–81% of all patients post-stroke, often developing during the early post-stroke stage, defined as 6 months after the cerebral insult [1,2,3]. Subluxation may cause shoulder pain, severely impact upper-limb function, constrain the performance of activities of daily living, and negatively affect the rehabilitation process beyond the acute stage [2,3,4,5,6]. Subluxation should be treated as early as possible as it can worsen over time and develop into an uncorrectable state [1]. Consequently, the management of this condition is believed to play an essential role in the rehabilitation process of patients post-stroke. Various approaches have been used to treat shoulder subluxation, i.e., anesthetic suprascapular nerve block, intramuscular injections of botulinum toxin, physical therapy, suprascapular nerve pulsed radiofrequency, corticosteroid injections, trigger-point dry needling, positioning, taping, mechanical supporting devices, and electrical stimulation [2,5].

Shoulder subluxation is the result of a decrease/absence in ability to resist the gravitational forces generated by the arm’s weight, particularly when the patient is in an upright position [7]. Stabilizing the humeral head in the glenoid fossa and counteracting this downward pull is achieved by the movement of the supraspinatus and posterior deltoid muscles. Weakness or paralysis of these muscles, as a result of the stroke, may result in inferior subluxation of the shoulder [1,4,8,9]. Accordingly, some treatment modalities aim to substitute the role of the paralyzed stabilizing muscle. These include the use of mechanical supportive devices (i.e., slings) or electrical stimulation of the supporting muscles.

Several systematic reviews and meta-analyses have reported the effectiveness of neuromuscular electrical stimulation (NMES) as a treatment modality for reducing shoulder subluxation in the early post-stroke stage [2,7,10,11,12]. However, many of these studies do not make a clear distinction between NMES and functional electrical stimulation (FES).

NMES delivers electrical stimulation via transcutaneous electrodes which are placed over the skin, covering the nerve innervating the paralyzed/weak muscle or the muscle belly without synchronization of the timing of the stimulation with spontaneous functional movement. FES is a subtype of NMES, in which stimulation is accurately synchronized with the voluntary contraction required to perform the desired movement, e.g., eliciting ankle dorsiflexion during the swing phase of gait. FES is usually applied in order to achieve specific functional and purposeful movements. The limitation of FES is the requirement of an expensive detection system, usually based on motion sensors, which enables accurate timing of the electrically induced contraction [13]. Conversely, NMES requires no feedback system and can be employed by most off-the-shelf electrical stimulation devices, providing that they include an on/off parameter. Its application is, therefore, simpler, cheaper, and more feasible. Yet, there is a need to investigate whether NMES applied during the performance of functional tasks and active motor learning, but not necessarily synchronized with the movements as FES, can benefit subjects with subluxation [2].

The role of mechanical supportive devices is to counteract the downward pull of gravity on the humeral head and ensure proper alignment of the scapular glenoid complex [12]. There are a variety of supportive devices in clinical use that include shoulder slings, orthoses, and wheelchair attachments [2,12,14]. Some systematic reviews have reported insufficient evidence from randomized controlled trials whether supportive devices indeed effectively prevent subluxation post-stroke [2,14,15]. Nonetheless, the advantages of supportive devices include their easy usability by caregivers and the possibility of applying these devices in conjunction with other treatment modalities [12]. Accordingly, it is reasonable to utilize the mechanical advantages of supportive devices in conjunction with the application of NMES during daily routine activities as a treatment modality for shoulder subluxation. To the best of our knowledge, only one parallel-group design study [16] explored the combination of NMES with shoulder support and support only in shoulder subluxation post-stroke. Although this study indicated a potentially positive effect on shoulder subluxation, its methodological quality was low (a 4 on the PEDro scale).

Therefore, the aim of this study was to explore the efficacy of long-duration NMES (up to 3 h a day) in conjunction with external support to the shoulder while the patient is engaged in a daily rehabilitation routine. The study examined the treatment effect on degree of subluxation, upper extremity motor function, daily functional ability, and shoulder pain. Our hypothesis was that the combination of shoulder external support and long-duration NMES during the daily rehabilitation routine would reduce the degree of subluxation and shoulder pain while improving upper-extremity motor function and daily functional ability more than shoulder external support alone.

## 2. Materials and Methods

### 2.1. Design

A prospective, parallel-group randomized controlled double-blind trial.

### 2.2. Sample Inclusion and Exclusion Criteria

Twenty-eight subjects with subluxation of the shoulder due to a stroke were recruited for this study. The inclusion criteria were as follows: acute phase of stroke (<6 months since cerebral insult), shoulder subluxation, and first stroke. The exclusion criteria were as follows: participation in other interventional clinical trials, ≤18 years of age, aphasia or cognitive disorders, inability to communicate with the research staff, history of severe health problems (i.e., other neurological, musculoskeletal, or mental disorders), and shoulder pain/trauma/operation in the relevant shoulder pre-stroke.

### 2.3. Sample Size Estimation

Sample size calculation was performed using Winpepi software, version 11.65, which was based on the outcomes of a previous trial that demonstrated a significant improvement in motor function, measured by the Fugl–Meyer motor assessment, following the application of NMES on the upper extremities in patients with stroke [17]. The calculation was based on the following assumption: a change of 11.9 in the Fugl–Meyer assessment-upper extremity (FMA-UE) score with a standard deviation of 3.95 in the experimental group and a change of 6.3 in the FMA-UE with a standard deviation of 4.6 in the controls with a difference of 5.6, with an α value of 5% and 80% power (probability). The required sample was 20 participants, with 10 in each group.

### 2.4. Recruitment and Randomization

Post-stroke patients were recruited from the Bait-Balev, Bat-Yam Rehabilitation Center, Israel. The participants were screened for inclusion and exclusion criteria by a certified occupational therapist. Individuals who met the required criteria and agreed to participate in the study, signed an informed consent form, and were randomly assigned to one of two groups using “Random Allocation software”.

### 2.5. Ethics Statement

The study was approved by the Helsinki Committee of Maccabi Healthcare’s Helsinki Committee, Israel (request code 0032-19-BB). The clinical trial was registered with the Israeli Ministry of Health-MOH 201-11-04-007470 on 27 October 2019. Consent for publication was obtained from all the patients, and the patients consented to having their photos published.

### 2.6. Intervention

An off-the-shelf NMES device (AS-TEC AS-1088-8, approved by the United States Food and Drug Administration) was used. The device program was set at a pulse duration of 250 μs, a pulse frequency of 35 Hz, and on/off times of 2.5 s/2.5 s. Pulse amplitude was set by the therapist to produce an observable motor response as recommended by the literature [10,12]. The electrodes were placed over the supraspinatus and the posterior deltoid muscles (Figure 1) [7]. The therapist ensured that the movement elicited by the stimulation was shoulder abduction and not shoulder elevation.

The experimental group received NMES treatment 5 days a week for 6 weeks. Treatment each morning consisted of three stimulation periods separated by 30 min rest intervals to allow adequate muscle recovery [18]. During the first week, each NMES period was 30 min long. Subsequently, the periods were gradually increased each week by 10 min up to a maximum of 60 min starting in the fourth week. Previous studies have found this protocol to be beneficial [7,10,19].

For the control group, the device was turned on and the stimulation parameters were adjusted, but the amplitude was not turned on. The subjects in this groups were told that they may or may not feel the stimulation. Treatment duration was identical in both groups.

External shoulder support (Figure 2) was individually adjusted to all patients who had undergone conventional therapy with an emphasis on shoulder strengthening.

Both groups continued their daily function and rehabilitation routine while using the NMES device. The subjects received only conventional treatment during the follow-up period (2 weeks after the completion of the 6 week treatment). Conventional therapy was provided by a therapist, blinded to group allocation.

### 2.7. Data Collection

Demographic data (age, sex, weight, and body mass index (BMI)), background diseases, depression, and stroke history (date of infarct, stroke type, area of stroke, affected side, and computed tomography results) were collected from the patients’ medical records. Sensory deficits were noted during the initial rehabilitation evaluation by the treating therapist. Outcome measures were evaluated three times during the study period: pre-intervention (T_0_), following 6 weeks of treatment (T_1_), and 2 weeks later as follow-up (T_2_). The assessments were performed by two certified occupational therapists with >3 years of experience in stroke rehabilitation, blinded to the participants’ group allocation. All the assessments were carried out by the same therapist, to ensure intra-rater reliability.

### 2.8. Outcome Measures

#### 2.8.1. Shoulder Subluxation Measurement

Shoulder subluxation measurement was determined by palpation of the gap between the acromion and the head of the humerus when the subject was seated without shoulder support [8]. To minimize the effect of normal anatomic variation, the affected shoulder subluxation was measured relative to the contralateral side [20], and the subluxation degree was measured by the fingerbreadth between the acromion and the head of the humerus [21]. This measurement is considered a useful clinical screening measure for shoulder subluxation [22] and is a valid tool significantly correlated with X-ray images (Spearman correlation, *r* = 0.76) [22], showing good inter-rater reliability (k = 0.900) [8] and intra-rater reliability (ICC = 0.90–0.94) [21].

#### 2.8.2. Upper-Extremity Motor Function

The Fugl–Meyer assessment-upper extremity (FMA-UE) scale assessed the sensory–motor impairments post-stroke. This commonly used clinical and research tool with excellent psychometrics evaluates the changes in motor impairments post-stroke [23], measuring movement, coordination, and reflex action. The FMA-UE consists of four subscales: shoulder, wrist, hand, and coordination/speed [24]. Each item is scored on a three-point ordinal scale: 0, unable to perform; 1, able to perform partially; 2, able to perform completely. A total maximum attainable score of 66 indicates normal motor functioning of the upper extremity [25].

#### 2.8.3. Distal Upper-Extremity Movement

The hand subscale of the FMA-UE assesses wrist stability and mobility, demonstrating that the hand subscale can be used as a valid and reliable “standalone” measure of distal upper extremity movement [26]. The subscale includes measurement of finger extension, finger flexion, and five types of grasps (hook, thumb, pincer, cylinder, and spherical). Rating is on a three-point ordinal scale: 0, unable to perform; 1, able to perform partially; and 2, able to perform completely. A maximal score of 14 indicates normal hand movement [26].

#### 2.8.4. Active Finger Extension

In order to measure the active finger extension, the item of “mass active extension” taken from the hand subscale of the FMA-UE was employed. A three-point ordinal scale was used: 0, unable to extend fingers to perform a full active or passive flexion; 1, partial finger extension; and 2, able to completely extend fingers.

#### 2.8.5. Pain Intensity

Pain intensity was evaluated by the Numerical Pain Rating Scale (NPRS), an 11-point scale graded from 0 (no pain) to 10 (worst pain) [27]. The scale is known to be a valid and reliable tool with a high correlation with the visual analog and verbal rating scales [28]. During the measurement, the subjects were asked to grade their pain intensity on average over the last 24 h [27]. Notably, the minimal clinically important difference (MCID) for shoulder pain was 2.17 [29].

#### 2.8.6. Daily Function

Daily function was evaluated using the Functional Independence Measure (FIM). The FIM measures self-care independence and is not specific to any pathology. The FIM instrument is an 18-item ordinal scale from 1 (complete dependence) to 7 (total independence) with scores ranging between 18 and 126 (low scores indicate lower independence in daily functioning) [30].

### 2.9. Statistical Analysis

SPSS software, version 23 (IBM Corp., Armonk, NY, USA) for Windows, was used for statistical analysis. Background characteristics between groups were compared using the Mann-Whitney and chi-square tests. Nonparametric tests were chosen to compare between groups due to the abnormal distribution of the data (according to the Q-Q plots). Outcome measures were compared between groups using the Wilcoxon two-sample test. In addition, three delta scores were calculated and compared: (1) post-intervention score minus pre-intervention score; (2) follow-up score minus post-intervention score; and (3) follow-up score minus pre-intervention score. Statistical significance was set at *p* < 0.05.

## 3. Results

Twenty-eight eligible patients participated in the study between 1 November 2019 and 15 March 2021 following the screening of 300 patients. The eligible patients were randomly allocated to two groups, with 14 patients in each group. Ten and thirteen participants of the experimental and control groups, respectively, completed the 6 week intervention. Four subjects dropped out of the experimental group as a result of the following reasons: one subject was hospitalized due to a deterioration in their medical condition, two were hospitalized due to the COVID-19 virus, and one participant disliked the stimulation sensation. In the control group, only one participant dropped out as he was hospitalized due to a deterioration in his medical condition. The follow-up assessment 2 weeks after the termination of the intervention period was conducted on eight and ten participants in the experimental and control groups, respectively. Five subjects dropped out at this stage (two in the experimental group and three in the control group) as they were discharged early from the rehabilitation center due to COVID-19 restrictions. A summary of the study progress is presented in Figure 3. No adverse events were recorded in any groups.

### 3.1. Background Characteristics of the Subjects per Group

The average age of the participants (±standard deviation (SD)) was 70.0 ± 14.0 years old, with an average time since stroke of 1.00 ± 1.4 months. Table 1 presents the background characteristics including the demographic and stroke characteristics of both groups. No statistical differences were found in the background characteristics of the participants (Table 1).

### 3.2. Outcome Measures

The outcome measures pre- and post-intervention, as well as at follow-up, per group and the statistical analysis results are presented in Table 2. No statistical differences were found between the study groups in all baseline outcome measures (Table 2). Comparisons of the three delta scores across groups and the statistical measures are presented in Table 3.

#### 3.2.1. Shoulder Subluxation

Subluxation measured in centimeters was significantly lower in the experimental group compared to the control group at both the post-intervention and the follow-up assessments (0.7 ± 0.82 vs. 2 ± 1.08, *p* = 0.0058; 0.38 ± 0.74 vs. 2.00 ± 1.20, *p* = 0.0045, respectively). The comparison of delta scores demonstrated a significant difference between groups only in the follow-up minus pre-intervention scores. When comparing the post-intervention minus pre-intervention assessment, a trend in favor of the experimental group was demonstrated.

#### 3.2.2. Motor Function

Participants in the experimental group showed significantly higher upper-extremity motor function (FMA-UE score) immediately following the intervention and at follow-up compared with the controls (44.7 ± 21.92 vs. 17.46 ± 16.00, *p* = 0.005; 51.00 ± 19.82 vs. 23.20 ± 17.55, *p* = 0.016, respectively). The comparison of delta scores demonstrated a significant difference between the groups in the post-intervention minus pre-intervention scores and in the follow-up minus pre-intervention scores, indicating the effectiveness of combining NMES with external shoulder support. No significant difference was demonstrated between the follow-up and post-intervention scores, demonstrating no deterioration in motor function when NMES was discontinued.

#### 3.2.3. Hand Movement

Participants in the experimental group showed a significant improvement in the FMA hand subscale, indicating more normal hand movement compared with the controls 6 weeks post-intervention and in the follow-up measurement (9.9 ± 5.32 vs. 3.15 ± 4.34, *p* = 0.006; 11.13 ± 4.73 vs. 4.40 ± 5.13, *p* = 0.030). When comparing the delta scores, a significant difference between groups was demonstrated only in the post-intervention minus pre-intervention scores (*p* = 0.014).

#### 3.2.4. Finger Extension

Participants in the experimental group showed a significant improvement in the FMA finger extension subscale, indicating more normal finger extension compared with the controls 6 weeks post-intervention and in the follow-up measurement (1.70 ± 0.67 vs. 0.62 ± 0.87, *p* = 0.007; 1.75 ± 0.71 vs. 0.90 ± 0.88; *p* = 0.038, respectively). When comparing the delta scores, a significant difference in the post-intervention minus pre-intervention scores (*p* = 0.014) was noted.

#### 3.2.5. Pain

There was no significant group difference in shoulder NPRS in the post-intervention and follow-up measurements. When comparing the delta scores, no significant difference between the groups was observed.

#### 3.2.6. Daily Living Function

No significant difference in the daily living function was observed in the FIM scores between the two study groups during the post-intervention and follow-up measurements. When comparing the delta scores between the follow-up and pre-intervention assessments, a significantly greater improvement at follow-up (*p* = 0.045) in the experimental group was observed. This improvement did not reach significance immediately post-intervention (*p* = 0.099).

## 4. Discussion

Shoulder subluxation is well documented in the literature as a predictor of rehabilitation outcomes following stroke [31,32]. Different approaches have been employed to improve shoulder subluxation and upper-extremity rehabilitation outcomes, including NMES, applied to the shoulder muscle, and different types of external shoulder support [12]. Nonetheless, there is no clear evidence as to the potential of using a combination of the two. We evaluated the potential of a therapy model that combines long-duration NMES and shoulder support while the patient is engaged in routine physical activities. Herein, we demonstrated that long-duration NMES vs. sham NMES, when applied with external shoulder support during a conventional rehabilitation program, significantly affected the overall upper-extremity motor function and distal hand movements. A trend was also shown regarding the effect of active NMES on shoulder subluxation and activities of daily living.

### 4.1. Shoulder Subluxation

Our results indicated that the addition of NMES to external shoulder support is beneficial in treating shoulder subluxation (absolute differences between groups in the post-intervention and follow-up assessments). The mean change in subluxation following intervention in the experimental group was 9 mm compared to 2 mm in the controls. These results are supported by a study [16] which examined the effect of NMES when supplemented with external shoulder support, and by a meta-analysis which examined the effect of NMES combined with conventional rehabilitation without external shoulder support. The mean reduction in shoulder subluxation in this meta-analysis was 6.5 mm following NMES, compared to 1.9 mm with conventional rehabilitation alone [10]. As the reduction in shoulder subluxation in the present study following NMES was higher, our findings may indicate the benefit of combining NMES with external shoulder support, although additional studies are necessary to substantiate this claim.

### 4.2. Motor Function

Our results demonstrated the beneficial effect of the combined therapy on upper-extremity motor function as reflected by the FMA-UE scores. A lack of previous studies regarding upper-extremity function impedes our ability to compare our data with the literature. However, this finding is consistent with Lin and Yan’s study [17], who demonstrated that 3 weeks of long-duration NMES as a supplement to conventional rehabilitation resulted in a significantly greater improvement in upper-extremity motor function in the early stages of stroke compared to conventional rehabilitation alone. However, whereas in the present study, the changes in the FMA-UE score were 20.0 points in the experimental group and 4.5 points in the controls, the change in the earlier study, where no external support was provided, was 11.9 in the experimental group and 6.3 in the controls [17]. These differences may be due to the longer duration of NMES (6 weeks compared to 3 weeks), not only because of the use of an external support [17]. It should be noted that the reported MCID for FMA-UE in patients with moderate to severe hemiparesis in the early stroke stage is 12.4 [33]. Our participants, with a baseline score of <35 on the FMA-UE, should be considered as having severe hemiparesis [33]. The change in our participants’ FMA-UE score following combination therapy (20.00) reached the MCID threshold, whereas the change in the controls of 4.46 was far from the MCID threshold. This finding emphasizes the importance of supplementing the rehabilitation regime with NMES in order to reach meaningful changes in motor function.

### 4.3. Hand Movement and Finger Extension

One of the main concerns post-stroke is the long-term hand dysfunction, which causes functional limitations with a prevalence of ~80% [34,35]. We demonstrated that the combined treatment had a significantly greater effect on hand function and finger extension following 6 weeks of intervention compared to the conventional intervention. The observed improvements in the hand and finger function may be due to the enhanced proximal stability offered by the treatment [36]. However, further studies are required, as the change scores in both groups did not reach the reported MCID for hand movement (Δ = 4.9) [33]. To the best of our knowledge, there are no reported MCID values of the finger extension subscale of the FMA—hand subscale.

### 4.4. Pain

Shoulder pain impeding rehabilitation is common in the early stage of stroke, with shoulder subluxation considered an etiological factor [37]. We found that neither NMES nor external shoulder support positively affected shoulder pain. Similarly, no effect on pain was observed in previous studies examining the effect of NMES and/or external shoulder support, despite a significant decrease in shoulder subluxation [2,14,16]. It is possible that shoulder pain following stroke is unrelated to shoulder subluxation but is related to other factors such as limited shoulder range of motion, muscle spasms, and central pain [37].

### 4.5. Daily Living Function

Despite treatment, no significant difference between groups was observed at each assessment session in daily living function, implying that there was no benefit to the combined treatment protocol. However, only in the experimental group did the overall change in daily living function at the follow-up assessment improve significantly compared to the pretreatment condition. A similar trend was observed in the post-intervention assessment. It is important to note that the magnitude of change post-intervention in the experimental group reached the MCID of FIM (>22) [38].

### 4.6. Follow-Up

It is important to note that all the post-intervention results were maintained at the 2 week follow-up assessment. Previous studies examining the effect of NMES alone on shoulder subluxation and motor function have presented controversial results as to the carry-over effect at follow-up [39,40,41]. In contrast to our results, a single report examining the effect of a combined treatment protocol (external shoulder support and NMES) specified that the treatment effect on shoulder subluxation was not maintained at follow-up [16]. While the difference may be due to the fact that our follow-up period was only 2 weeks compared to their 3 months follow up, it is also possible that the difference was related to the intensity and duration of treatment. The variables which may enhance treatment benefits over extended periods of time warrant further investigation.

### 4.7. Study Limitations

The present study’s results were limited to patients at the acute stage following a stroke and cannot be generalized to patients in the more chronic stages. A methodological issue that should be considered is that, in the majority of the cited studies, shoulder subluxation was measured by X-ray imaging. In contrast, in the current study, the degree of subluxation was determined by palpation due to the limited availability of radiographic equipment. Whereas radiological images are most probably more accurate, the intra-rater reliability of palpation has been shown to be excellent (ICC 0.9–0.94) [21] and to correlate well with radiological findings (*r* = 0.8) [22]. Since each of our patients was assessed by a single therapist blinded to group allocation, we believe that our results are valid. On the other hand, using less sophisticated instruments such as palpation can serve as a quick, feasible, and cost-effective assessment method in a clinical setting, as suggested in the field of pulmonary rehabilitation [42]. Similar to most previous studies, the baseline magnitude of subluxation was not deemed as a criterion for inclusion. This factor should be further examined in future studies as it may affect the results. Further studies may expand the understanding of the mechanism of the two treatment modalities for shoulder subluxation by measuring motor activity in the shoulder muscle using EMG assessments [43,44]. Similar to most previous studies, the baseline magnitude of subluxation was not deemed as a criterion of inclusion. This factor should be further examined in future studies as it may have affected the results. Furthermore, NMES was applied only during the morning hours due to clinical constraints. It is possible that longer treatment duration would have been more beneficial. Lastly, a follow-up assessment was carried out 2 weeks post-treatment. Longer follow-up assessments are necessary.

## 5. Conclusions

We demonstrated that a combination of long-duration NMES and external shoulder support during the routine rehabilitation of individuals at the acute stage post-stroke is an effective modality for improving distal hand movements and upper-extremity motor function. A trend in favor of the combined intervention was shown to reduce shoulder subluxation and improve daily living function. All of the observed changes were maintained at the 2 week follow-up assessment.

## Figures and Tables

**Figure 1 ijerph-19-09765-f001:**
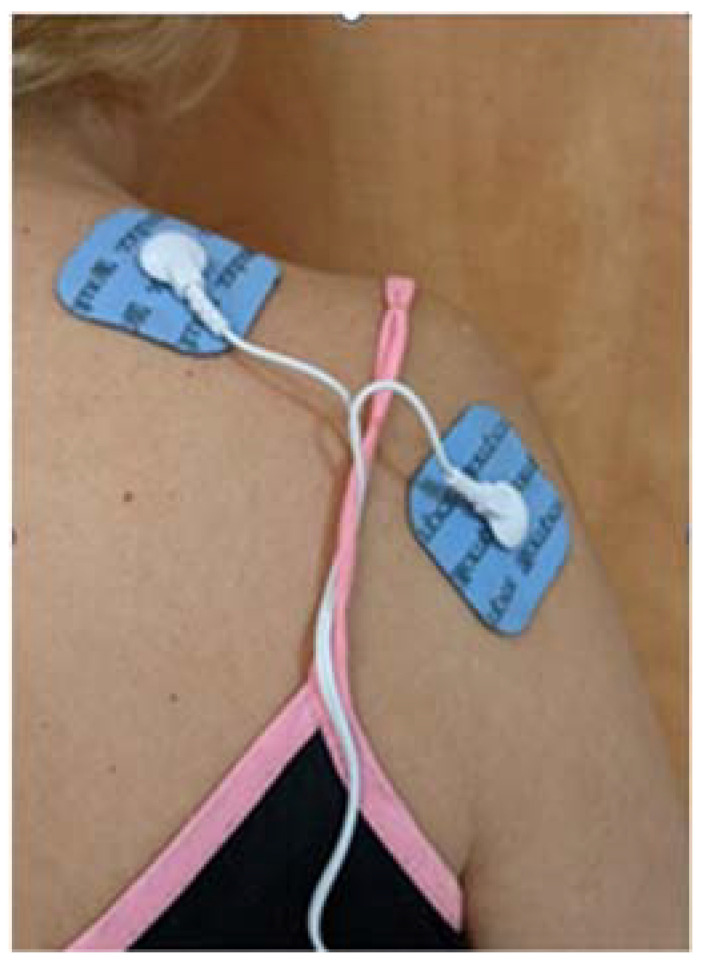
Example of FES electrode placement.

**Figure 2 ijerph-19-09765-f002:**
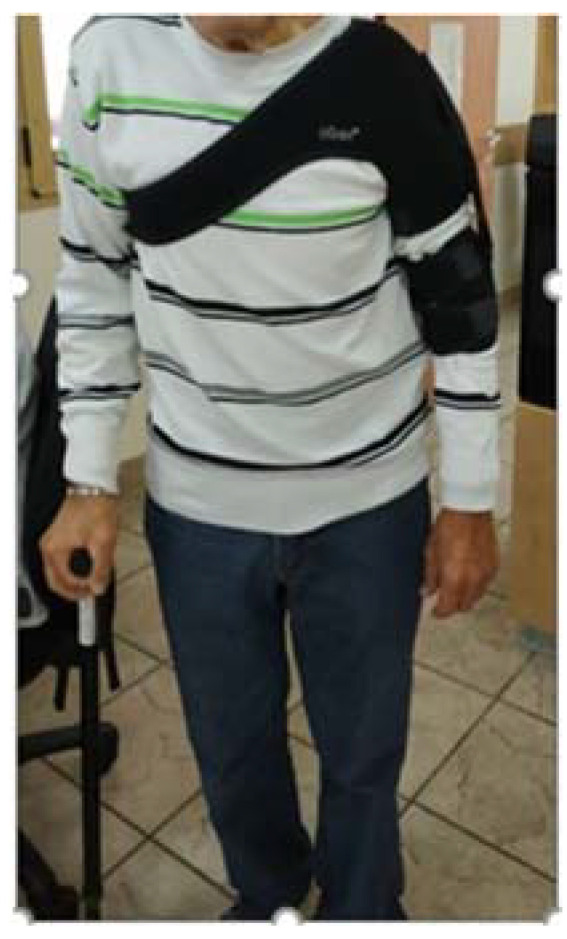
Example of shoulder support.

**Figure 3 ijerph-19-09765-f003:**
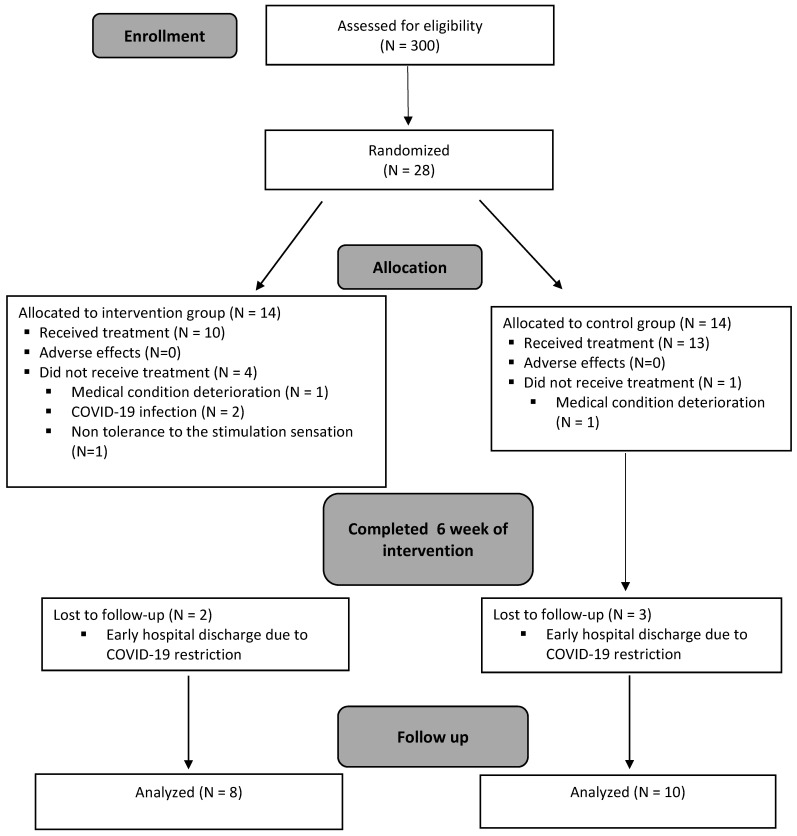
Flowchart of the study design.

**Table 1 ijerph-19-09765-t001:** Background characteristics (demographic and stroke characteristics) of the subjects per group.

Variables	Control Group *n* = 13 (Mean ± SD)	Experimental Group *n* = 10 (Mean ± SD)	Mann-Whitney (U, *p*-Value)
Age (years)	67.54 ± 15.54	73.30 ± 9.81	U = 48, *p* = 0.306
Weight (kg)	70.97 ± 12.02	65.75 ± 11.01	U = 50, *p* = 0.352
BMI (kg/m^2^)	25.72 ± 4.87	25.23 ± 3.26	U = 47, *p* = 0.849
Time since stroke (months)	1.38 ± 1.61	0.50 ± 0.97	U = 45, *p* = 0.169
**Variables**	**Control Group**	**Experimental Group**	**Chi-Square χ^2^ (*p*-Value)**
Sex	Male 61.5% Female 38.5%	Male 60% Female 40%	Fisher, *p* = 1
Background disease	Yes 69.2% No 30.8%	Yes 70% No 30%	Fisher, *p* = 1
Type of stroke	Ischemic 76.9% Hemorrhagic 23.1%	Ischemic 80% Hemorrhagic 20%	Fisher, *p* = 1
Affected side	Right 53.8% Left 46.2%	Right 50% Left 50%	Fisher, *p* = 1

SD-Standard Deviation; BMI-Body Mass Index. Statistical significance was set at *p* < 0.05.

**Table 2 ijerph-19-09765-t002:** Outcome measures pre- and post-intervention following 6 weeks of treatment and at follow-up 2 weeks later per group (mean ± standard deviation, (median)) and Wilcoxon two-sample test results.

	Pre-Intervention (T_0_)			Post-Intervention Following 6 Weeks of Treatment (T_1_)			Follow-Up 2 Weeks Later (T_2_)		
	Experimental *n* = 10	Control *n* = 13	*p*-Value	Experimental *n* = 10	Control *n* = 13	*p*-Value	Experimental *n* = 8	Control *n* = 10	*p*-Value
Shoulder subluxation (cm)	1.60 ± 0.84 (2.00)	2.23 ± 0.93 (2.00)	0.148	0.70 ± 0.82 (0.50)	2.00 ± 1.08 (2.00)	0.0058	0.38 ± 0.74 (0.00)	2.00 ± 1.20 (2.10)	0.0045
* FMA-UE (0–66)	24.70 ± 17.98 (30.00)	13.00 ± 11.80 (4.00)	0.099	44.70 ± 21.92 (53.50)	17.46 ± 16.00 (15.00)	0.005	51.00 ± 19.82 (55.00)	23.20 ± 17.55 (22.50)	0.016
Hand, FMA-UE (0–14)	5.20 ± 4.83 (5.00)	2.00 ± 3.08 (0.00)	0.094	9.90 ± 5.32 (11.50)	3.15 ± 4.34 (0.00)	0.006	11.13 ± 4.73 (13.00)	4.40 ± 5.13 (3.00)	0.03
Finger Extension (0–2)	0.90 ± 0.99 (0.50)	0.46 ± 0.78 (0.00)	0.285	1.70 ± 0.67 (2.00)	0.62 ± 0.87 (0.00)	0.007	1.75 ± 0.71 (2.00)	0.90 ± 0.88 (1.00)	0.038
NPRS (0–10)	4.30 ± 3.80 (4.00)	3.92 ± 3.28 (4.00)	0.825	4.00 ± 3.20 (5.00)	3.46 ± 2.54 (4.00)	0.639	2.88 ± 2.75 (2.00)	3.60 ± 3.24 (4.00)	0.786
FIM (18–126)	58.30 ± 15.46 (57.50)	52.00 ± 22.35 (44.00)	0.456	81.20 ± 21.16 (83.00)	63.00 ± 27.40 (59.00)	0.172	89.50 ± 22.33 (94.50)	69.90 ± 27.10 (60.00)	0.168

* FMA-UE: Fugl–Meyer assessment—upper extremity; NPRS: Numerical Pain Rating Scale; FIM: Functional Independence Measure. Statistical significance was set at *p* < 0.05.

**Table 3 ijerph-19-09765-t003:** Comparison of the delta scores (post-intervention score minus pre-intervention score (T_1_ − T_0_), follow-up score minus post-intervention score (T_2_ − T_1_), follow-up score minus pre-intervention score (T_2_ − T_0_) between groups; mean ± standard deviation (median)) and Wilcoxon two-sample test results.

	Post-Intervention Minus Pre Intervention			Follow-Up Minus Post Intervention			Follow-Up Minus Pre Intervention		
	Experimental *n* = 10	Control *n* = 13	*p*-Value	Experimental *n* = 8	Control *n* = 10	*p*-Value	Experimental *n* = 8	Control *n* = 10	*p*-Value
Shoulder subluxation	−0.90 ± 1.20 (−1.00)	−0.23 ± 0.60 (0.00)	0.0964	−0.13 ± 0.35 (0.00)	0.30 ± 0.67 (0.00)	0.1158	−1.38 ± 0.92 (−1.00)	0.00 ± 1.05 (0.00)	0.0107
* FMA-UE	20.00 ± 20.09 (15.50)	4.46 ± 12.31 (0.00)	0.006	0.75 ± 1.67 (0.50)	1.50 ± 3.21 (0.00)	1.0000	24.88 ± 20.51 (23.50)	7.50 ± 16.30 (0.50)	0.035
Hand, FMA-UE (0–14)	4.70 ± 4.95 (2.00)	1.15 ± 3.34 (0.00)	0.014	0.00 ± 0.53 (0.00)	0.30 ± 1.25 (0.00)	0.49	5.50 ± 5.21 (4.50)	1.80 ± 4.89 (0.50)	0.1040
Finger extension (0–2)	0.80 ± 0.92 (0.50)	0.15 ± 0.55 (0.00)	0.036	0.00 ± 0.00 (0.00)	0.10 ± 0.32 (0.00)	0.434	0.88 ± 0.99 (0.50)	0.30 ± 0.67 (0.00)	0.182
NPRS (0–10)	−0.30 ± 4.11 (0.00)	−0.46 ± 4.03 (0.00)	0.826	−0.25 ± 1.39 (0.00)	0.10 ± 3.54 (0.00)	0.854	−1.38 ± 4.07 (−0.50)	−1.30 ± 4.92 (−1.00)	0.964
FIM (18–126)	22.90 ± 17.50 (21.00)	11.00 ± 11.02 (6.00)	0.099	8.63 ± 8.58 (5.50)	1.70 ± 2.54 (0.00)	0.062	31.88 ± 16.48 (34.00)	14.90 ± 13.22 (14.00	0.045

* FMA-UE: Fugl–Meyer assessment-upper extremity; NPRS: Numerical Pain Rating Scale; FIM: Functional Independence Measure. Statistical significance was set at *p* < 0.05

## Data Availability

Data are available on request from the authors.

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
