# Peer review of "A Combination of Long-Duration Electrical Stimulation with External Shoulder Support during Routine Daily Activities in Patients with Post-Hemiplegic Shoulder Subluxation: A Randomized Controlled Study"

_ijerph, 2022, doi:10.3390/ijerph19159765_

Round 1
Reviewer 1 Report
The study aim was to determine the effect of long-duration neuromuscular electric stimulation (NMES applied to the supraspinatus and posterior deltoid muscles on the paretic side) on shoulder subluxation and the upper extremity function during the acute post-stroke stage when additionally the standard rehabilitation (not precised in details) and external shoulder support (EST) were applied. In fact, although easy to understand from the experimental design reasons, the ethical aspects of this study is questionable because the other sham group received only rehabilitation and support (?). What were criteria of such a choice (not mentioned in the text) of medical treatment except the randomization of the study? More detailed flow chart is necessary in the text, to understand all applied principles.
The study was performed on the small number of participants (twenty-eight subjects) versus similar sham group with mean age ± standard deviation until 70.0±14.0 years, what really makes the confidence of the applied clinical tests for the paretic function evaluation disputable (degree of shoulder subluxation; Fugl-Meyer assessment upper extremity test (FMA-UE); FMA hand and finger subscales; Functional Independence Measure (FIM); and shoulder pain (with Numeric Pain Rate Scale), because they rely on the confidence of the patients report and their mental (with the full respect) abilities.
Some aspects of the study design should be explained. First the shoulder subluxation and upper extremity function pathology during the acute post-stroke stage can be two different issues. They can have their pathological origin either after the acute stroke or independently following the mechanical injury, authors did not clarified them in the text, and the proposed treatment might have different influence to these two pathologies of the different origin (if such a situation had happened). Are the authors really sure that the subluxation was/is only the consequence of the stroke?
Authors claim (lines 19-20) in the Abstract that…” the changes in the subluxation, FMA-UE, hand and finger FMA subscales were significantly higher in the experimental group post-intervention and at follow-up.”… Subluxation higher? What did they take the conclusion from that (lines 20-21) …”Supplementing NMES to standard rehabilitation and EST is beneficial in reducing shoulder subluxation and improving upper extremity function.”… . ?
The Introduction section is written convincing, contrary, in the aim and Material and Methods sections includes data which is surprising. The experimental group received NMES treatment five days a week for six weeks (long enough to verify the long-term effect of stimulation), up to 3 hours a day (!, which is extremely long in duration, not previously described in the literature, contrary to the authors description in lines 99-102), with the strength of applied stimuli (not precised in mA) up to the visible motor response (muscle’s stretch? but which one?), if the monopolar stimulation comprised both muscle belly of more trapezius than supraspinatus and deltoid muscles, as it has been presented in Fig. 1 for electrodes placement. This photograph (in Fig. 1) presents the electrode location on the trapezius muscle belly more than supraspinatus one. Authors didn’t mention in the text that they stimulated simultaneously two muscles' bellies (both trapezius and deltoids). Moreover, do authors realize what happens with the activity of the muscle motor units overloaded with the electrical stimulation for more than 3 hours a day? Such a prolonged stimulation would bring side effects. Was it reported by the patients? This is the major concern of the presented study.
Results are presented well enough, Discussion is very concise as well as the Conclusions.
The paper needs some editorial improvements like big and small letters, commas, dots application etc. (for example line 13, comma before bracket not necessary, many other similar are included in the article content including different font sizes), English grammar mistakes corrections. It seems that the article content was prepared in the great rush, including the horizontal position of the text layout what makes the small letters reading irritable.
Author Response
We thank you and the reviewers for the thoughtful suggestions and insights. The manuscript has benefited from these insightful suggestions. We look forward to working with you and the reviewers to move this manuscript closer to publication in IJERPH.
The manuscript has been rechecked and the necessary changes have been made in accordance with the reviewers’ comments and suggestions. In addition, language editing was done by MDPI language editing Services.
The responses to all comments have been prepared and attached herewith below.
The changes made are highlighted in the attached revised manuscript.
We hope this will enable you to accept our paper for publication in your esteemed journal.

Reviewer 2 Report
Reviews
Introduction
Kindly make your second paragraph the first paragraph as it is defining what is shoulder subluxation.
In lines 44-46, The NMES lacks timing synchronization with functional movement, such is not the case for FES. This makes FES a better choice, but despite this property, NMES is cheaper and requires no feedback. Thus making it a better choice for rehabilitation (here). It is important to note that a clear organized and contrasting statement like these will create a more clear story. Therefore, I suggest rewriting this paragraph as recommended and creating a more clear story.
Paragraph 51-60 is not clear, Are you saying that NMES is generally used in conjunction with mechanical supportive devices? You are testing long-term usage of NMES with supportive devices on the post-stroke shoulder during a functional task. If you can clarify this in the first few lines or beginning of the paragraph that NMES + Support device + functional task are used for rehabilitation, it will be better. Add some references regarding the combination of NMES, therapy, and supportive device to pace up rehabilitation. It is clear to me that a person who is not familiar with our area will have a tough time understanding that you are using a combination of functional tasks and supportive devices with NMES to prevent subluxation. I suggest clarifying this paragraph more.
Overall, I liked the introduction. It is well written with the research gap identified and how your study fulfills it. I recommend adding your hypothesis in the introduction.
Materials and Methods
Kindly mention the specific name of the power test that determined the sample size of your study.
Kindly provide the rationale why pulse duration and frequencies were set at such parameters (line 92).
Figures 1 and 2 should be merged and organized well. Kindly check some other papers for more organized figures. Also, add captions below the figure explaining what these images mean.
My question regarding NMES is how it was used as a placebo even though you are still triggering pulses of a certain duration with an off amplitude. I am not able to understand how can you trigger a pulse of zero amplitude. Kindly clarify this.
Moreover, if you are using a constant amplitude pulse for a placebo, is not it possible that a pulse of a certain duration will be having some impact on the result. Hence, it is not completely a placebo. Kindly provide a reference around this argument.
Results
I suggest using the word sex rather than gender as it will clarify the question that the participants were biologically male and biologically female.
I like the flow chart of the study.
What is the background characteristic? Kindly clarify this term in the text. I can understand it but clarify it, such as what factors are associated with background characteristics. It will provide more clarity to the readers as background characteristics are a very broad term.
It will be convenient if you define post-intervention, preintervention, and other terms used at the top of the tables.
I like the way you have summarised your results for the motor function subsection (lines 213-214). It will be better if you provide such a single statement summary for other sub-sections in the results. In short, what does this significant difference means in terms of recovery of the finger, hand, and other movements?
Discussion
The discussion is written well. However, some other instrumentations can record and assess motor impairment in shoulder muscles. I recommend citing some literature regarding EMG assessments. I am suggesting a few below for ease [1], [2].
Conclusions
The conclusion is well written
References
1. Abd AT, Singh RE, Iqbal K, White G. Investigation of Power Specific Motor Primitives in an Upper Limb Rotational Motion. J Mot Behav. 2022;54(1):80-91. doi: 10.1080/00222895.2021.1916424. Epub 2021 Jun 24. PMID: 34167442.
2. Roh J, Rymer WZ, Perreault EJ, Yoo SB, Beer RF. Alterations in upper limb muscle synergy structure in chronic stroke survivors. J Neurophysiol. 2013 Feb;109(3):768-81. doi: 10.1152/jn.00670.2012. Epub 2012 Nov 14. PMID: 23155178; PMCID: PMC3567389.
Author Response

(The authors gave the same response as above.)

Reviewer 3 Report
The authors have developed an interesting and useful retrospective study on the effect of long-duration neuromuscular electric stimulation on shoulder subluxation and upper extremity function during the acute post-stroke stage.
However, I would like to make some observations before recommending your work for publication.
1. Please improve the visibility of the Flow Chart (Figure 3).
2. Please remove the contours of Figure 1 and 2.
3. Line 116 of section "2.7 Data Collection", delete the "etc...", and detail in full
4. While the measurement of subluxation by palpation may appear to be a limitation, it is actually a strength because clinicians often need quick and cost-effective measurements. In this sense, I recommend the authors to mention this in the "Discussion" section, and also to mention the use of low-cost measurement tools in other pathologies, such as COPD, citing the following work: DOI: 10.29333/ejgm/11671
5. In line 308 of "Coclusions", remove the "f".
Author Response
01/08/2022
We thank you and the reviewers for the thoughtful suggestions and insights. The manuscript has benefited from these insightful suggestions. We look forward to working with you and the reviewers to move this manuscript closer to publication in IJERPH.
The manuscript has been rechecked and the necessary changes have been made in accordance with the reviewers’ comments and suggestions. In addition, language editing was done by MDPI language editing Services.
The responses to all comments have been prepared and attached herewith below.
The changes made are highlighted in the attached revised manuscript.
We hope this will enable you to accept our paper for publication in your esteemed journal.
Sincerely,

Round 2
Reviewer 1 Report
First, I would like to congratulate the authors. The quality of the paper has greatly improved. They modified the text describing the study not only following my suggestions but also, I suppose, thanks to the indications of other reviewers, fully answering my concerns as follows:
-They modified the Intro section with valuable data and developed the hypothesis
-They modified results description in the abstract, now it is more clear
They introduced the additional data on the patients selection in M&M section
-They explained the ethical considerations, rules pf participation by subjects in each of the studied groups.
-They explained criteria of medical treatment in groups of patients as well as the randomization of the study. They provided more details in the text and modified the flowchart.
-They explained the choice and reliability of applied methods of patients evaluation, mentioning in the Study Limitations the future necessity of electromyography (EMG) utilization
-They explained the shoulder subluxation origin among patients in the text
-The provided more references explaining the choice of applied electrotherapy principles
-They explained lack of side effects following the applied electrotherapy
-They almost convincingly explained the rules of bipolar stimulation, with observations of no overlapping stretch responses from both supraspinatus and trapezius muscles, although the simultaneous stimulations of these two muscles group with one train of electrical pulses don’t seem for me selective. However, the authors relied on other descriptions of such stimulation, providing suitable refs.
-They performed some editorial improvements and necessary English editing making the paper better than the primary version
I believe that in the current form the paper is suitable for printing in IJERPH
Reviewer 3 Report
The authors have improved the previous version of their manuscript, and it is now ready for publication.
Congratulations